# Stabilization of Anthocyanins from Coffee (*Coffea arabica* L.) Husks and In Vivo Evaluation of Their Antioxidant Activity

**DOI:** 10.3390/molecules28031353

**Published:** 2023-01-31

**Authors:** José Daniel Lozada-Ramírez, María Cristina Guerrero-Moras, Marco Antonio González-Peña, Taisa Sabrina Silva-Pereira, Cecilia Anaya de Parrodi, Ana E. Ortega-Regules

**Affiliations:** 1Department of Chemical and Biological Sciences, Universidad de las Américas Puebla, 72810 San Andrés Cholula, Puebla, Mexico; 2Department of Health Sciences, Universidad de las Américas Puebla, 72810 San Andrés Cholula, Puebla, Mexico; 3Department of Chemical, Food and Environmental Engineering, Universidad de las Américas Puebla, 72810 San Andrés Cholula, Puebla, Mexico

**Keywords:** *Coffea arabica*, anthocyanin extraction, anthocyanin stabilization, zinc oxide nanoparticles, in vivo antioxidant activity, *Caenorhabditis elegans*

## Abstract

Coffee (*Coffea arabica* L.) is one of the most popular and widely consumed products throughout the world, mainly due to its taste, aroma, caffeine content, and natural antioxidants. Among those antioxidants, anthocyanins are one of the most important natural pigments, which can be found in coffee husks. It is widely known that anthocyanins have multiple health benefits partially linked to their antioxidant properties. However, anthocyanins have low stability and are sensitive to all types of changes. In order to prevent its degradation, anthocyanins can be stabilized with nanoparticles. Thus, the main objective of this study was to evaluate the stability of the anthocyanins extracted from coffee husks, using three different extracting agents (ethanol, methanol, and water) and stabilizing them through conjugation with zinc oxide nanoparticles. The anthocyanins extracts were mainly composed of cyanidin-3-rutinoside (97%) and the total phenolic compounds of the fresh extracts were 458.97 ± 11.32 (methanol), 373.53 ± 12.74 (ethanol), and 369.85 ± 15.93 (water) mg GAE/g. On the other hand, the total phenolic compounds of the nanoparticle–anthocyanin conjugates underwent no significant changes after stabilization as the major loss was less than 3%. Furthermore, the percentage of anthocyanins’ degradation was less than 5% after 12 weeks of storage. On top of that, fresh anthocyanin extracts and anthocyanin–nanoparticle conjugates exhibited a strong protective effect against oxidative stress and increased the survival rate of *Caenorhabditis elegans*.

## 1. Introduction

Coffee (*Coffea arabica* L.) is one of the most consumed products around the globe and is highly appreciated for its taste, aroma, and caffeine content [1]. From 2016–2017, the world’s production of coffee was 9.1 thousand million kilograms, increasing to 10.1 and 10.2 thousand million kilograms in 2017–2018 and 2018–2019, respectively, thus demonstrating its economic and commercial importance around the world [1,2,3]. The grain of the fruit of coffee is the main-processed product from the plant, and it is necessary for it to be separated from the surrounding pulp. This process, known as coffee pulping, generates large amounts of organic residues that must be treated in order to not contaminate the environment [4]. Favorably, this organic matter can be used as a source of biogas and other biofuels including, ethanol and solid biofuel, which are highly appreciated for diverse purposes [5,6]. On the other hand, the peel of the fruit of coffee is rich in polyphenols, bioactive compounds with numerous properties beneficial to human health [7]. A kind of polyphenols present in coffee husks, among other kinds of compounds, are anthocyanins, one of the most important natural pigments the human eye can recognize; their color variation goes from vibrant red to different types of purple and blue [5,7,8]. Anthocyanins have a basic structure C6C3C6 as the rest of the flavonoids, albeit anthocyanins, have a particular ability to form the flavylium cation, which provides them with their unique positive charge [9,10,11]. Their basic structure is an anthocyanidin, which is an aglycone, with at least one glucoside substitute [11,12].

The main interest in studying these pigments is their multiple health benefits. It has been proven that anthocyanins have anti-inflammatory effects for different diseases, such as colitis, reflux, and pain diminution, among others [13]. This anti-inflammatory response is due to the blocking of the expression of the transcription factor NF-κB for the synthesis of pro-inflammatory enzymes (COX-2) and the MAPK activation pathway [14,15]. In addition, anthocyanins play a protective role in cardiovascular diseases through the increment of PPAR and the downregulation of CD36 expression to prevent platelet aggregation and LDL oxidation; moreover, anthocyanins increase polyunsaturated fatty acid levels over saturated fatty acid levels [12,16,17,18]. Furthermore, these pigments have hypoglycemic and anti-obesity properties by increasing GLUT4 expression through AMPK activation leading to better glucose catchment by muscles and fat tissue [19,20]. Furthermore, anthocyanins decrease PEPCK expression and upregulate PPAR, CPT1A, and ACO, providing a reduction in lipid liver levels [21,22]. Moreover, anthocyanins have neuroprotective effects in Parkinson’s disease and Alzheimer’s disease through the activation of FKBPs and a reduction in TAU protein levels. Moreover, anthocyanins decrease Ca^2+^ intracellular levels and enhance neurons myelination [23,24]. Aspects that stand out for anthocyanins are their antioxidant properties, by far the most important benefit, and as a result, their ability to prevent or inhibit molecular oxidation by neutralizing free radicals. Their antioxidant capacity depends on the ring orientation and the –OH substituents. These pigments inhibit cancer cell proliferation and induce apoptosis. It has been proven that anthocyanins provoke apoptosis in CaCo-2 cells, diminish Hep3B cells’ migration by decreasing phosphorylation of GSK3β, and β-catenin expression. In addition, these wild pigments regulate proteins such as *p-53* and *c-myc* that participate in different inflammation and carcinogenesis signaling cascades [17,25,26,27,28].

Because of all their health benefits and their bright colors, anthocyanins can be used as natural colorants as well as functional foods. Coffee peels are rich in anthocyanins, particularly cyanidin-3-rutinoside, which is one of the most active anthocyanins, acting as a powerful antioxidant [5,8,29]. Nevertheless, their low stability makes them sensitive to all types of changes, including pH, temperature, light, enzymes, oxygen, and so forth. At a pH just under 3, anthocyanins are in their most stable forms. Between pH 3–6, the flavylium cation is lost; as a result, these pigments become non-colored. At an alkaline pH, anthocyanins are in their most unstable forms; at this point, their positive charges become negative, which could lead to irreversible damage [30,31,32]. In the same way, above 40 °C, the glucoside substituent is lost, creating an unstable base [33,34,35,36]. Additionally, at higher water activity, there is more interaction between water and the flavylium cation, leading once again to an unstable base [37,38,39].

Nanoparticles may stabilize anthocyanins preventing their degradation due to their physical, biological, and chemical unique properties; nanoparticles can be used as a bio-encapsulant agent for drugs and bioactive substances, improving their bioavailability [40,41,42,43,44,45]. In addition, nanoparticles are useful in the prevention of some cancers, enhancing the permeability of bioactive substances [46,47,48]. In addition, the biological model *Caenorhabditis elegans* has been used as a research model for understanding the metabolic, pathological, and molecular mechanisms associated with the aging process, the development of diseases, the function, antioxidant capacity, and toxicity of foods, bioactive compounds, and plant extracts [49,50]. For the aforementioned reasons, *C. elegans* has become a model to evaluate the augmented lifespan and/or resistance against oxidative stress when exposed to natural antioxidants, with the advantage to evaluate the effect across multiple generations due to its capacity to produce multiple descendants [51,52].

ZnO has been extensively used in the biomedical, pharmaceutical, cosmetic, and dental industries, but it has also been explored for food applications. In the case of the nanoparticles of ZnO, because they are more soluble than bulk ZnO, they can be used as biocompatible materials, with the additional advantage that they are cheap and can be easily synthesized [53,54,55,56,57]. It has been observed that ZnO is less toxic to biological systems, with Zn^2+^ being released from this material, a trace element in biological systems, thus making this compound biocompatible [58,59]. Nevertheless, ZnO is not completely inert because it has small but controlled chemical reactivity [60].

The main objective of this study was to evaluate the stability of anthocyanins extracted from coffee husks, using ZnO nanoparticles as immobilizing agents, following the time evolution related to their antioxidant capacity and phenolics concentration. Characterization of the ZnO nanoparticle–anthocyanins preparations was evaluated, demonstrating a stabilization process because of the formation of those conjugates. The antioxidant effect was also tested in vivo using the biological model *C. elegans*, comparing the protective effect of the nanoparticle–anthocyanin conjugates in an oxidative environment versus the fresh extract of anthocyanins.

## 2. Results and Discussion

### 2.1. Extraction and Phenolic Compounds’ Determination

Three different extracting agents were used (MeOH, EtOH, and water) to extract anthocyanins from 30 g of coffee husks. The results indicate that the organic extracting agents had a better performance than water, especially MeOH. MeOH extracts presented more phenolic compounds than the other extracts (*p* < 0.05), with 458.97 ± 11.32 mg of GAE/g obtained, in comparison to the 373.53 ± 12.74 mg GAE/g and 369.85 ± 15.93 mg GAE/g obtained from EtOH and water, respectively (Figure 1). The extraction using methanol reached 20% higher yields of phenolic compounds than EtOH and water, which relates to previous reports [61].

Other studies focused on measuring the total phenolic compounds’ concentration of fruits of the berry family [62] reported that blackcurrant (*Ribes nigrum*) had the highest values of phenolic compounds, followed by blackberry (*Rubus ulmifolius*) and strawberry (*Fragaria sevatis*). The Appendix A summarizes the values of phenolic compounds and monomeric anthocyanins and the antioxidant activity of some reported berries and other fruits [63,64].

### 2.2. Anthocyanins’ Determination

The anthocyanin profile was obtained by HPLC at a wavelength of 520 nm (Figure 2A). The spectrum showed two peaks at retention times of 13.8 and 15.8 min, representing 11% and 89%, respectively. These results agree with previous reports [8,65,66]. Those peaks correspond to cyanidin 3-glucoside and cyanidin 3-rutinoside, with the wider peak and the most anthocyanin extracted being shown for the latter. The spectrum also shows a high purity of the anthocyanins. Cyanidin 3-rutinoside is also the majority of the anthocyanin in lychee with a 97% composition [67,68,69]. Additionally, for the identification of the anthocyanins extracted, they were applied to NMR spectroscopy (Figure 2B). The H^1^ spectrum shows two singlets at 8.421 ppm and at 7.090 ppm with a J of 1.365 Hz and 1.509 Hz, respectively. In addition, two doublets were found at 7.808 and 7.789 ppm and at 6.280–6.284 corresponding to 2′, 5′, and 3′ protons and to 8′ and 6′protons, respectively. A triplet can be found at 6.780 ppm. These results correspond to the characteristic signals to the cyanidin aglycone [5,66], confirming that cyanidin-3-rutinoside is the most extracted anthocyanin from the coffee cherry.

The vibrational frequencies obtained by FTIR spectroscopy can be observed in Figure 3; the graphs show a well-defined peak at 1000 cm^−1^ and these peaks correspond to the anthocyanins [70,71]. After the stabilization with ZnO nanoparticles, most of the peaks diminish or disappear, suggesting an interaction between the nanoparticles and the anthocyanin extracts. The interaction changes the usual vibrational frequencies of the functional groups; however, the anthocyanins’ reference peak at 1000 cm^−1^ remains with no apparent changes in all of the graphs [71]. According to previous reports [72], ZnO can bond with –OH functional groups by forming hydrogen bonds between the zinc and the –OH in the positions R1, R2, R5, or R7. Because of the electrostatic nature of the bonds, their bonding energies are week, which may help the anthocyanin to adopt a non-stressed conformation and just a dislocation of the electronic density. To evaluate the stabilization of the anthocyanins, the phenolic compounds and the antioxidant activity were measured over the time.

The AAS analysis was performed after the dissolution of ZnO nanoparticles with the extracts of anthocyanins. Our results provide evidence of stabilization of anthocyanin mediated by uncoated nanoparticles dissolving more than three-fold when maintained for 120 h in solution, compared to the ZnO–anthocyanin preparation (Figure 4). The fast release of Zn^2+^ from ZnO was evidenced from the beginning of this study, reaching a loss of 54.9%. On the other hand, the sample prepared with anthocyanins attached to nanostructured ZnO evidenced a loss of 17.3%. These findings confirm the stabilization of the nanoparticles in the solution.

### 2.3. Measurements of Stabilized Nanoparticles

The total phenol compounds for the fresh anthocyanins were 458.97 ± 11.32 mg GAE/g for the methanol extract, 373.53 ± 12.74 mg GAE/g for the ethanol extract, and 369.85 ± 15.93 mg GAE/g for the water extract (Figure 1). After the stabilization with the ZnO nanoparticles, the variation in the values of the total phenol compounds did not represent a significant change (*p* > 0.05); the graph comparing the initial values of the un-stabilized anthocyanin extract and the stabilized anthocyanin extract can be found in Figure 1. ZnO stabilization should not affect the phenol compounds in the extracts, and, as the values confirmed, the stabilization did not significantly change the values in comparison to the unstabilized ones (*p* > 0.05). Water-stabilized anthocyanins presented the most radical decrease of all the extract agents used in this study. However, that change was from 369.85 ± 15.93 mg GAE/g to 359.43 ± 10.21 mg GAE/g. The difference of 10 mg GAE/g represents a decrease of 2.82% which can be considered negligible (*p* > 0.05). After 12 weeks of measurements, the values of the phenolic compounds were 452.87 ± 12.62 mg GAE/g, 356.31 ± 11.29 mg GAE/g, and 354.37 ± 7.3 mg GAE/g for the methanol, ethanol, and water extracts, respectively. Those differences exhibit a decrease of 1.33%, 4.61%, and 4.18%, respectively, after a period of 12 weeks. These results suggest that in 12 weeks, the percentage of anthocyanins degradation was less than 5%. Santos and Gonçalves [73] reported degradation of the phenolic compounds in ethanolic, methanolic, and water extracts from flour prepared by using a mix of selected fruits, with values reaching higher than 77% of loss after 25 weeks of storage and more than 50% after 10 weeks.

Prata and Oliveira [8] and Murthy et al. [66] reported the presence of cyanidin 3-glucoside and cyanidin 3-rutinoside as monomeric anthocyanins in coffee cherry, in quantities of 19 and 25 mg C3G/100 g sample, respectively. In our study, values of 16.7 (MeOH), 14.9 (EtOH), and 12.3 (water) mg C3G/100 g sample were achieved. Some authors have previously addressed the monomeric anthocyanins’ content of various fruit, which are described in the Appendix A. Similar to those observed for phenolic compounds, the methanol extract exhibits a higher content of monomeric anthocyanins compared to the ethanol extract (7%) and to the water extract (25%).

For the ZnO-stabilized anthocyanins, the value of 14.9 mg C3G/100 g sample was achieved for the methanol extract. This value represents 89% of the retained anthocyanins on the nanomaterial, compared with the original methanolic extract. Similarly, 11.0 mg C3G/100 g sample was measured when the ethanol extract was stabilized with ZnO nanoparticles (73.8% retention) (Figure 1).

The initial values of antioxidant activity for the anthocyanins’ extracts, without stabilization, were 1576 µmol eq. Trolox for the methanol extract; 1309 µmol eq. Trolox for the ethanol extract; and 1057 µmol eq. Trolox for the water extract. The methanol extract presented the highest antioxidant activity at the beginning (*p* < 0.05).

Additionally, it also showed the highest value of phenolic compounds (*p* < 0.05), which is clearly related. This is because the more anthocyanins that are present, the more radicals can stabilize, thus, increasing the antioxidant activity. The initial values of the antioxidant activity of the fresh anthocyanin extract and the values of the stabilized anthocyanin extract were compared, and this comparison is shown in Figure 1. The anthocyanins in methanol exhibited a minor loss effect after stabilizing the extracts with ZnO nanoparticles, displaying a 5.3% decrease in antioxidant activity after the stabilization, while the water extract demonstrated major loss after the stabilization (*p* < 0.05) with a decrease of 22.6%. For the antioxidant activity measurements of the anthocyanins stabilized with ZnO, monitoring for a period of 12 weeks was carried out. ZnO nanoparticles without the anthocyanin extract were used as controls, and zinc ions from ZnCl_2_ were evaluated to measure if the ZnO nanoparticles or the Zn^2+^ ions had any effect on the antioxidant activity by themselves. The process to determine the antioxidant activity of the ZnO nanoparticles and Zn^2+^ ions was exactly the same as the one described in the Materials and Methods, with the ABTS+ radical. Both, ZnO nanoparticles and Zn^2+^ ions did not present any antioxidant activity, as expected, meaning that they do not directly influence the antioxidant activity. After an evaluation period of 12 weeks, the antioxidant activity of the stabilized anthocyanins decreased more than the total phenolic compounds. Again, the water-extracted anthocyanins presented the most prominent decrease in antioxidant activity (*p* < 0.05), which corresponds to a 58.2% reduction. That variation is attributed to the easy degradation process that the anthocyanins may suffer during the 12 weeks [74]. Although the loss in antioxidant activity of the water-extracted anthocyanins may seem like a considerable loss, the normal antioxidant activity of the anthocyanins in usual conditions does not exceed two weeks, with some exceptions [75], and at 12 weeks, our stabilized anthocyanins still preserved almost 50% of their activity, while for the methanol and ethanol extracted anthocyanins, the values of the antioxidant activity were considerably higher. For the methanol-extracted anthocyanins, the final value after 12 weeks was 1352 µmol eq. Trolox, and 973 µmol eq. Trolox for the ethanol-extracted anthocyanins; both values correspond to decreases of 14.2% and 25.7% in antioxidant activity, respectively. Figure 1 shows those losses in antioxidant activity regarding time.

As for visual effects, neither a precipitate after the whole 12 weeks nor a significant change in the coloration were observed. The anthocyanins also served to “stabilize” the ZnO nanoparticles in polar solvents, meaning that the anthocyanins kept the ZnO nanoparticles without aggregation, which is a highly desired characteristic for several applications, such as electronic applications and in particular biomedical applications [55,56,57,58]. Since anthocyanins are antioxidants with highly biocompatible and non-toxic compounds, they can also decrease the toxicity of the bare ZnO nanoparticles or other kinds of nanoparticles [56,59], which is a highly debatable topic.

### 2.4. Biological Assay

A helpful approach to evaluate the protective effect in oxidative stress is the use of nematode *C. elegans* to evaluate its adaptive capacity to those stressful conditions, which can be enhanced by the use of antioxidant molecules. Phenolic compounds as part of a person’s diet are beneficial to human health due to their antioxidant capacities [76,77,78]. Anthocyanins have beneficial effects due to their property of neutralizing free radicals, which are responsible for cell damage and complications in different pathologies related to oxidative stress. Anthocyanins maintain normal vascular permeability, show anti-inflammatory response protection against cancer, and prevent neuronal damage during aging [75,78]. Coffee is rich in antioxidant compounds; particularly, coffee cherry is considered as an excellent source of anthocyanins because of richness in cyanidin-3-rutinoside [8]. For the oxidative stress resistance assay, 10 mM H_2_O_2_ was used; it is important to mention that only the ethanol-extracted anthocyanins and the nanoparticles attached to anthocyanins from the same extract were used for the biological assay; the above is due to the fact that methanol is highly toxic to biological systems [79] and the stabilized water extracts exhibit low antioxidant capacity. Two assays of response to oxidative stress were performed in nematodes from P_0_ to F_4_ generation in the larvae stage L4, which were treated with anthocyanins extract from coffee husks and with ZnO nanoparticles attached to anthocyanins from the same extract. Both treatments were compared versus the control group, which did not receive any treatment. The nematodes were transferred to plates with pro-oxidant conditions (H_2_O_2_) and their vitality was tested each hour. Once individuals received the treatment with the ethanol extract of anthocyanins or with the ZnO nanoparticles attached to anthocyanins extracted with ethanol, they were transferred to plates with H_2_O_2_, and their mobility was analyzed until the control group was entirely dead. The number of living individuals on the treated experiments varies along the five generations of worms tested. During the generation P_0_, the entire control group died after 7 h of the initial exposition to H_2_O_2_, meanwhile 4% of the group treated with the anthocyanins extract were alive after 9 h (*p* < 0.05). The group treated with ZnO nanoparticles with anthocyanins vary related to the group treated with a fresh extract, presenting with 11% survival after 9 h (*p* < 0.10). For the generation F_1_, none of the worms survived the exposition of H_2_O_2_ after 9 h of treatment, which is contrary to the 16% and 17.7% (*p* < 0.05) of survival of the worms treated with anthocyanins in extract or attached to ZnO nanoparticles, respectively. For the case of the F_2_ and F_3_ generations, similar behavior was observed, whereby the control group in both generations showed a survival rate of 0.33% and 0%, respectively. The worms treated with the anthocyanin extract registered a survival rate of 13.7% (F_2_) (*p* < 0.05) and 17% (F_3_) (*p* < 0.05), and for those treated with anthocyanins attached to ZnO nanoparticles, the survival rates were 30.3% (*p* < 0.05) and 31.3% (*p* < 0.05), respectively. Finally, for the F_4_ generation, similar behavior was observed; 1% of the worms survived in the control group, 26% (*p* < 0.05) survived for the anthocyanin extract, and 30% (*p* < 0.05) of the individuals survived for the experiments using ZnO nanoparticles attached to coffee anthocyanins (Figure 5).

When the treatments were compared, there were not any significant differences observed among the treatments (*p* > 0.05). Nevertheless, a protective effect is evident in the nematodes treated with fresh anthocyanins and with ZnO nanoparticles with anthocyanins attached. To demonstrate the lack of an effect assigned to ZnO nanoparticles, an experiment was carried out using ZnO nanoparticles as antioxidant agents, prepared as described in Materials and Methods.

According to these results, we can affirm a beneficial and protective effect of the anthocyanins extracted from coffee husks versus oxidative stress, with an increase in the resistance of the nematodes being evident in both of the two treatments for each generation when compared to the controls. The protective effect could be demonstrated, but no significant statistical differences were found between the treatments (*p* > 0.05).

## 3. Materials and Methods

### 3.1. Materials

Coffee (*Coffea arabica* L.) cherry was acquired from “El Dos” Ranch, Atotocoyan, Yaonáhuac, Puebla in Mexico. Zinc chloride from Sigma Aldrich (Toluca, Mexico), hydrochloric acid (37%) from Sigma Aldrich (Toluca, Mexico), sodium hydroxide from Sigma Aldrich (Toluca, Mexico), potassium chloride from Sigma Aldrich (Toluca, Mexico), acetic acid from Sigma Aldrich (Toluca, Mexico), sodium acetate from Sigma Aldrich (Toluca, Mexico), potassium persulfate from Sigma Aldrich (Toluca, Mexico), ethanol (99.8%) from Sigma Aldrich (Toluca, Mexico), methanol (99.8%) from Sigma Aldrich (Toluca, Mexico), 6-hydroxy-2,5,7,8-tetramethylchroman-2-carboxylic acid (Trolox) from Sigma Aldrich (Toluca, Mexico) and 2,2′-azinobis-(3-ethylbenzothiazoline-6-sulfonic acid (ABTS) were purchased from Sigma Aldrich (Toluca, Mexico). Folin–Ciocalteu reagent was purchased from Hycel (Zapopan, Mexico). All chemical reagents were used without further purification.

### 3.2. Anthocyanin Extraction

The extraction of anthocyanins from coffee (*Coffea arabica*) husks was based on the previously reported methodology [66,80]. Briefly, 600 mL of three different extracting agents, water, methanol, or ethanol, were added to 30 g of coffee husks and left stirring for 2 h in the absence of light. For the case of anthocyanin extraction only using water as an extracting solvent, 30 g of coffee husks were ultrasonicated with distilled water for 10 min. The supernatant of all of the samples was centrifuged at 6000 rpm, then filtered to remove any solid material from the coffee skins and concentrated using a rotary evaporator Büchi 461 (Flawil, Switzerland) with a vacuum pressure of 56 cm of Hg at 35 °C, until a 35.5% concentration of soluble solids was reached. The extracts were stored frozen in darkness and hermetically sealed until further use (Appendix A).

### 3.3. Anthocyanin Purification

In order to get rid of all other undesirable components that may also be extracted during the procedure described previously, the extracts were purified by column chromatography using Amberlite XAD7HP from Sigma Aldrich (Toluca, Mexico) as the stationary phase. The purification process was performed as reported elsewhere [30,81]. Acidic methanol (0.01% *v/v* HCl) was used as eluent phase. The extracts were washed three times with distilled water and acidic methanol. Lastly, the anthocyanin extracts were redispersed in their respective solvents, i.e., methanol, ethanol, and water.

### 3.4. Anthocyanin Separation and Identification

To determine the anthocyanins extracted, HPLC Waters 600, Agilent (Santa Clara, CA, USA) coupled with a diode array detector was applied. The extract obtained from coffee peels was analyzed by HPLC equipped with a 20 μL loop, using a reverse phase column C18 LiChroCART (25 × 0.4 cm, 5 μm) Merck (Darmstadt, Germany). A gradient was established using acetonitrile (A) and 4.5% formic acid (B): isocratic 9:91 (A:B) for 25 min, 26–28 min, 100:0 (A:B) and 28–30 min, 9:91 (A:B) at a flow rate of 1.5 mL/min. Detection was carried out using a Waters 996 PDA detector at 520 nm [66,82,83].

The NMR spectrum was acquired using an NMR Varian Gemini 2000 (Palo Alto, CA, USA) spectrometer at 200 MHz, with a pulse sequence configuration for H^1^, using 5 mm sample tubes. The extract was diluted in deuterochloroform solution with TMS as the standard for the proton reference frequency (δ = 0) [66,84]. The FTIR spectra for the anthocyanins with and without a stabilizing agent were acquired using a Cary 60 Agilent Technologies (Tokyo, Japan) by measuring from 4000 cm^−1^ to 500 cm^−1^ [70,71].

### 3.5. Quantification of Total Phenolic Compounds

The total phenolic compounds of the anthocyanin extracts were determined using the Folin–Ciocalteu method. The Folin–Ciocalteu regent is a combination of phosphomolybdic acid and phosphotungstic acid, which reduces at contact with phenol compounds resulting in an intense blue coloration. For the determination, 7.5 mL of distilled water was added to 1 mL of the diluted extract (1:100) and 300 µL of the diluted Folin–Ciocalteu reagent (1:1 *v/v*). After 3 min, 1 mL of sodium carbonate (20% *w/v*) was added in order to neutralize organic acids. The measures were acquired 20 min after the addition of sodium carbonate, using a Shimadzu UV-1800 spectrophotometer (Tokyo, Japan) with a measuring wavelength of 760 nm. The absorbance of all the samples was compared to a previously prepared calibration curve of gallic acid and the results were expressed in milligrams of gallic acid per gram of sample (mg GAE/g) [85,86].

### 3.6. Total Monomeric Anthocyanins

Determination of total monomeric anthocyanins was performed following the pH differential methodology, described by Giusti and Wrolstad [87]. Briefly, the method is based on the shift of the anthocyanin structure in spite of pH, from intense red at pH 1 (flavylium cation) to colorless at pH 4.5 (hemiacetal). A 10 μL aliquot of sample was diluted to 1 mL with buffer pH 1 (HCl/KCl 0.025 M) and left to stand for 15 min. After that, the resulting sample was spectrophotometrically measured at 520 nm, ensuring the absorbance of the sample was on the 0.200–1.200 interval. The sample was set to pH 4.5 using 0.4 M acetic acid/sodium acetate buffer, measuring the absorbance at 700 nm and at λ_max_. The final absorbance (A_final_) was calculated using the following equation:A_final_ = (A_max vis_ − A_700nm_)_pH1.0_ − (A_max vis_ − A_700nm_)_pH4.5_

The A_final_ value was substituted in the following equation to obtain the anthocyanin concentration in the sample:Monomeric anthocyanins (mg/L) = (A_final_) (MW) (DF) (1000)/(ε)
where MW is the molecular weight of the anthocyanin most common in nature (cyanidin 3-glucoside, 449.2 g/mol) [88], DF is the dilution factor of the sample, and ε the coefficient of molar absorptivity (26,900 L/mol·cm).

### 3.7. Antioxidant Activity

The antioxidant activity was measured following the ABTS method [89], based on the capacity of the sample to catch the ABTS^+^ radicals; this can be monitored by measuring the fading of the color, which is proportional to the antioxidant activity. The ABTS^+^ radical was produced by adding a solution of 7 mM of ABTS with a 2.45 mM solution of K_2_S_2_O_8_ and it was left incubating for 16 to 24 h at room temperature in darkness. The ABTS^+^ solution was diluted with ethanol until an absorbance of 0.70 ± 0.02 was reached with a 754 nm wavelength. The extract was diluted with ethanol by adding 990 µL of ethanol to 10 µL of the anthocyanin extract to obtain an inhibition of 20–80% compared to the absorbance of the ABTS^+^ in ethanol. A solution containing 980 µL of the diluted ABTS^+^ was mixed with 20 µL of the diluted anthocyanin extract and left for 7 min until the absorbance was measured. To quantify the inhibition percentage, the following equation was used:$$Inhibition\;percentage=\left(\frac{Abs_{ini}-Abs_{final}}{Abs_{ini}}\right)\left(100\right)$$
where *Abs_ini_* is the initial absorbance and *Abs_final_* is the final absorbance. The reference antioxidant used was Trolox (6-hydroxy-2,5,7,8-tetramethylchroman-2-carboxylic acid), and the results were based on a Trolox calibration and expressed in units of Trolox micromoles equivalent (µmol eq Trolox).

### 3.8. Preparation of ZnO Nanoparticles

Zinc oxide nanoparticles were prepared following our previous methodology [72]. Briefly, a solution of 0.10 M of ZnCl_2_ in ethanol was added dropwise to a 0.10 M solution of NaOH in ethanol. The mixture was left stirring for 2 h and then washed three times by centrifugation using ethanol to remove the excess of ions.

### 3.9. Anthocyanin Stabilization Using ZnO Nanoparticles

To stabilize the anthocyanins extracted from coffee husks with the prepared ZnO nanoparticles, 7 mL of the purified anthocyanin extract in their extracting agent (methanol, ethanol, or water) was added to previously synthetized ZnO nanoparticles and left stirring for 48 h in darkness. After stabilization, the anthocyanins in the ethanol changed their color, from an intense red coloration to an intense blue, but when the pH was stabilized with HCl, the red coloration was recovered; in addition, the water-extracted anthocyanins changed from a red coloration into dark brown; the methanol-extracted anthocyanins kept their initial red coloration after stabilization. In order to evaluate the stabilization of the anthocyanins by the ZnO nanoparticles, a determination of total phenolic compounds, total monomeric anthocyanins, and antioxidant activity were measured over a period of 12 weeks [72].

### 3.10. Quantification of Anthocyanins Stabilized with ZnO Nanoparticles

To determine the mass of the anthocyanins coated by the ZnO nanoparticles, atomic absorption spectrometry Varian SpectrAA 220FS (Palo Alto, CA, USA) was used. For the determination, 1 mL of the stabilized anthocyanins with the ZnO nanoparticles was dried and dissolved completely by adding 1 mL of concentrated hydrochloric acid for 1 h. After the nanoparticles were completely dissolved, they were filled with distilled water until 1 L was reached and then measured with the instrument. The intensity of the signal obtained by the atomic absorption spectrometer was compared to a calibration curve previously made, to convert the intensity into the mass of the zinc ions. After the mass of the zinc ions was obtained, the total mass of ZnO was calculated according to stoichiometry and subtracted to the mass that was dried before, in order to obtain the milligrams of anthocyanins coating the nanoparticles [72].

### 3.11. Caenorhabditis Elegans Culture, Maintenance, and Syncronization

The nematode *Caenorhabditis elegans* (strain N2, Bristol wild type) [90] was grown to maintain a continuous supply using M9 buffer and NGM prepared as previously described [91,92,93]. *Escherichia coli* strain OP50 serves as food for *C. elegans*; this was cultured in LBM until it was seeded on NMG plates for *C. elegans* intake. *C. elegans* was maintained in NGM agar plates and transferred to fresh plates every 5–7 days by washing the nematodes with M9 buffer, following the methodology proposed by Stiernagle [91]. It was necessary to synchronize nematodes to the same larvae stage to make comparable observations. To synchronize *C. elegans*, parental nematodes were grown on 10-cm NGM plates and observed under a microscope to count the number of individuals with the same size and characteristics (100 individuals). Synchronization was realized using a alkaline hypochlorite solution. This treatment is based on the fact that gravid adults are sensitive to the bleach solution and when they become in contact with it, they dissolve, while embryos are protected by the eggshell [94]. When enough parental nematodes could be observed under the microscope, the NGM plates with *C. elegans* were washed with M9 buffer and put into Eppendorf tubes and centrifuged for 1 min at 4600 rpm (4 °C). The worms in the pellet were washed with 1 mL of NaOH, vortexed for 30 s, and centrifuged for 30 s at 4000 rpm (4 °C). The supernatant was removed and the worms were washed with 1 mL of sodium hypochlorite 5%:1 M NaOH (2:3, *v:v*), vortexed for 1 min, and centrifuged under the same conditions. Finally, the embryos were washed 2–3 times with 1 mL M9 buffer and then transferred onto NGM agar plates with *E. coli* OP50.

### 3.12. Biological Assay

To determine the antioxidant effect of the anthocyanins extracts, and to compare the efficiency of the stabilized anthocyanins with ZnO nanoparticles against the pure extract, the nematode *C. elegans* was exposed to a severe treatment of oxidative stress using hydrogen peroxide (H_2_O_2_). For that purpose, a dietary supplementation of *C. elegans* with anthocyanins was planned. Since *E. coli* OP50 is the only food source for *C. elegans*, any treatment addressed to *C. elegans* will be absorbed via *E. coli*. Therefore, dietary supplementation must be addressed to *E. coli* [95]. The biological assay consisted in the growth of *C. elegans* eggs in the following manner: (a) control, NGM with *E. coli*; (b) NGM with *E. coli* supplemented with a 50 µM fresh extract of anthocyanins of each solvent (water, ethanol, and methanol; (c) NGM with *E. coli* supplemented with 50 a µM ZnO–anthocyanin complex obtained from each solvent. *C. elegans* was cultured under those conditions for 48 h, then transferred to new plates (100 nematodes per plate in triplicate) prepared with the oxidant agent (10 mM H_2_O_2_). Survival was tested each hour and the nematodes were counted as being dead when no response was observed to a stimulus caused with a platinum handle [96,97]. To determine the effects of the treatment in the second generation, the treated nematodes were synchronized, and the eggs obtained were plated with antioxidants (fresh extract of anthocyanins and a nanoparticle–anthocyanin complex) until L4 phase was reached. The survival was evaluated following the methodology above. This procedure was repeated until the oxidative stress resistance of five generations was evaluated.

### 3.13. Statistical Analysis

To analyze the obtained results, the statistical software Minitab 18th version (State College, PA, USA) and GraphPad Prism 6.0 (San Diego, CA, USA) were used. The resistance to oxidative stress was analyzed using the differential proportions test, and the treatments were analyzed through the ANOVA test.

## 4. Conclusions

Polar organic solvents such as methanol and ethanol are good alternatives to extract anthocyanins from coffee husks. However, water is not the best extracting agent, since it may cause faster oxidation and decomposition of the anthocyanins, not mentioning that the number of phenolic compounds extracted with water was not as good as the number of phenolic compounds extracted with methanol and ethanol.

ZnO nanoparticles where synthetized through an easy methodology, followed by easier anthocyanin stabilization. Although the anthocyanin stabilization methodology did not present much difficulty, the results obtained were better than expected, because at 12 weeks, in the best scenario, the anthocyanins only lost 14% of their antioxidant activity and less than 5% of the phenolic compounds. The interactions between the ZnO nanoparticles and the anthocyanins extracted from coffee husks may explain these results. We believe that the attachment is caused by purely weak electrostatic interactions that help anthocyanins to organize and probably re-organize, as they need to entering a less energetic state. This can be achieved by a different polarization of the electronic density of the anthocyanins.

Anthocyanins are a very promising coating, since they are easily extracted from vegetables and fruits and do not present any toxicity risk. Therefore, it is possible to use these preparations for biomedical applications, since they can reduce the toxicity caused by the use of other materials. Further work regarding the stabilization of anthocyanins with other nanoparticles is currently being carried out in order to better understand the interactions formed by ZnO and anthocyanins. The positive antioxidant effect of the anthocyanins’ ethanolic extract, free or bonded to nanoparticles, observed in five generations of *C. elegans* cultures can be attributed to the positive effect against strong oxidative conditions of these natural pigments.

The findings of this study demonstrate the positive antioxidant effect of anthocyanins from coffee husks, a by-product of the food industry. Additionally, the high stabilization of these bioactive compounds has been achieved, shown by the low stability of free anthocyanins. The addition of nanostructured ZnO for the stabilization of these antioxidants has no effect on their optical properties nor on their beneficial effects, as evidenced in biological trials on *C. elegans*. Nevertheless, we can mention that more studies must be performed to evaluate the application of these preparations to human illnesses and that these preparations are limited to low-scale syntheses. On the other hand, it is possible to study the in vivo performance of free and stabilized anthocyanins from coffee, over human cancer lines cells or the murine model. In addition, it is possible to synthesize large amounts of ZnO nanoparticles to prepare conjugates with anthocyanins that can be useful for the pharmaceutical, biomedical, and cosmetic industries. For all of the above, we can affirm that ZnO nanoparticles are useful in stabilizing anthocyanins for several weeks and that these compounds can be applied to evaluate their beneficial properties in biological systems.

## Figures and Tables

**Figure 1 molecules-28-01353-f001:**
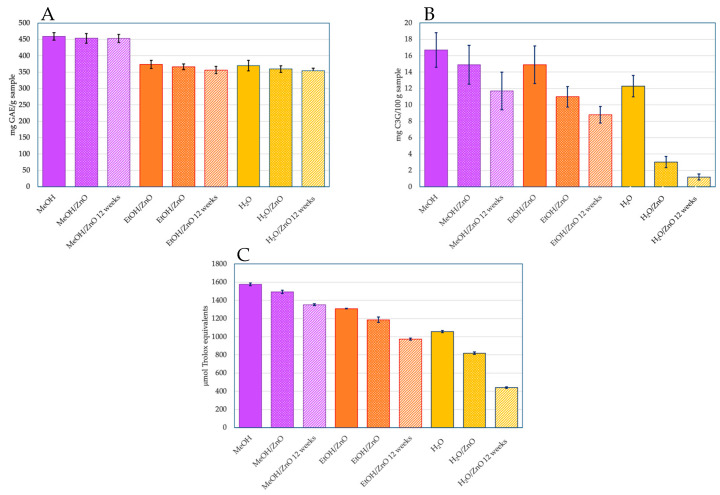
Comparison of total phenolic compounds (**A**), total monomeric anthocyanins (**B**), and antioxidant activity (**C**) of anthocyanins extracted from coffee (*Coffea arabica* L.) husks with MeOH, EtOH, or water (H_2_O) versus anthocyanins attached to zinc oxide nanoparticles (initially and after 12 weeks). The data shown correspond to means ± standard deviations (n = 3).

**Figure 2 molecules-28-01353-f002:**
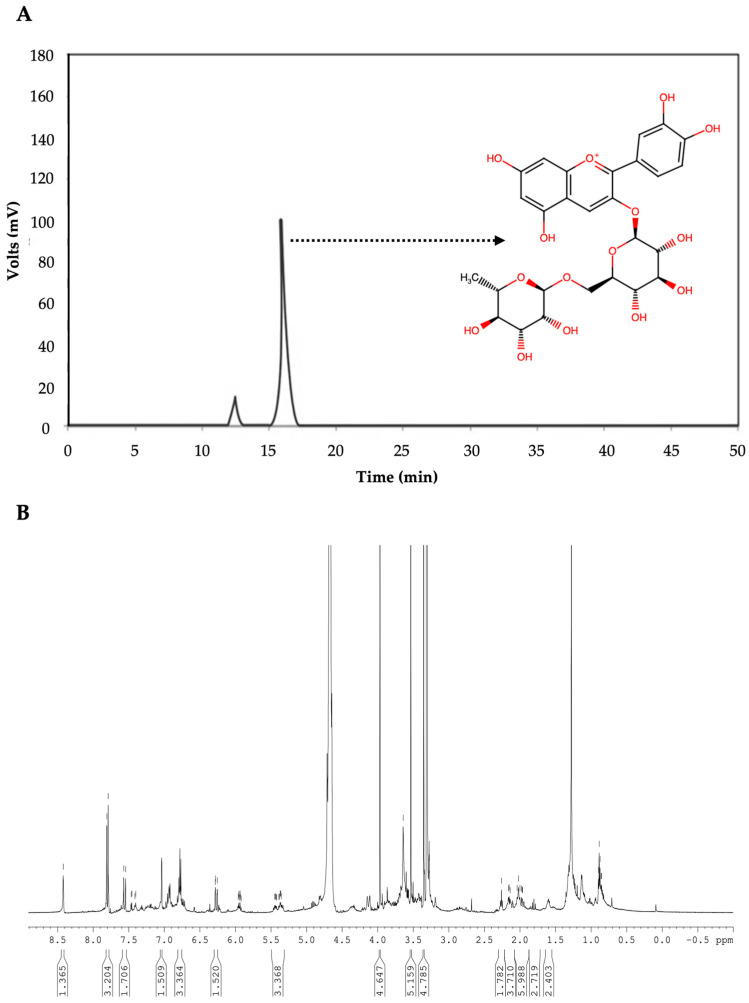
(**A**) HPLC profile of coffee (*Coffea arabica* L.) husk extracts. The two peaks in the HPLC profile correspond to cyanidin 3-glucoside (13.8 min) and cyanidin 3-rutinoside (15.8 min). (**B**) The H^1^ spectrum of the fraction obtained from the chromatographic separation of cyanidin 3-rutinoside. Refer to the text for the explanation.

**Figure 3 molecules-28-01353-f003:**
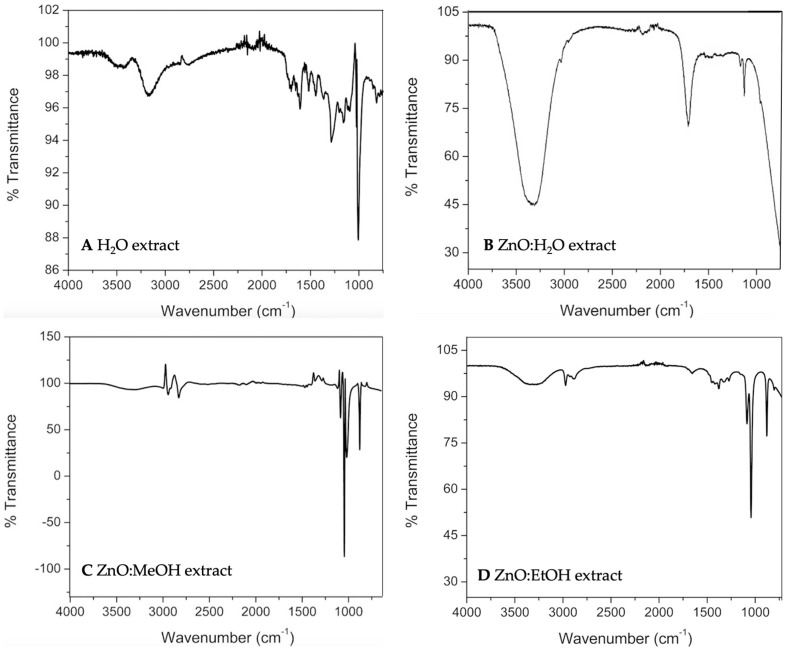
FTIR spectra for the water extract of coffee (*Coffea arabica* L.) husks (**A**). A characteristic signal for cyanidin 3-rutinoside appears at 1000 cm^−1^. ZnO nanoparticle–anthocyanin conjugates from water (**B**), MeOH (**C**), and EtOH extraction (**D**). The presence of the cyanidin 3-rutinoside’s signal is still present after the immobilization in all of the cases.

**Figure 4 molecules-28-01353-f004:**
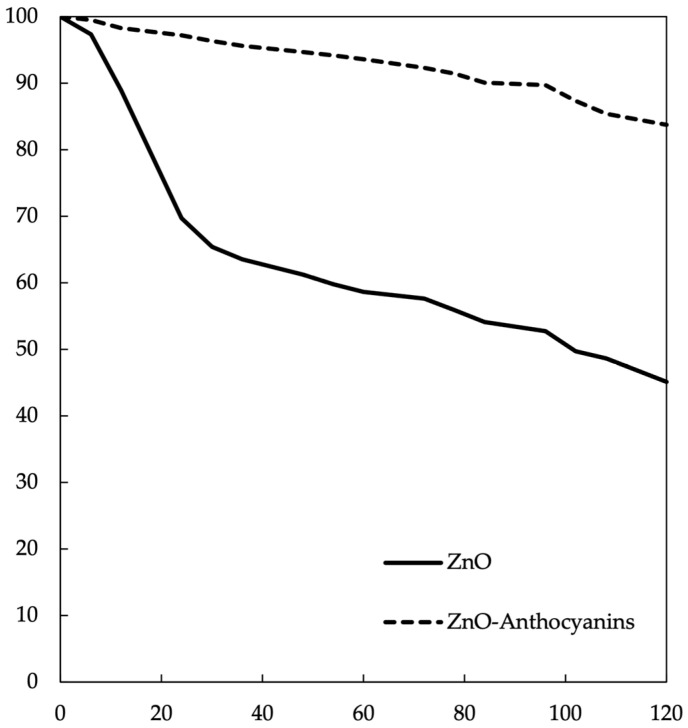
The maximal concentration of Zn^2+^ was quantified by digesting 10 mg of ZnO or ZnO–anthocyanin in HNO_3_ (concentrated) for 2 h and measuring the concentration of Zn^2+^ by AAS. The Zn^2+^ concentration was calculated using a calibration curve prepared previously (Appendix A) and the measures were performed at pH 7 and at room temperature.

**Figure 5 molecules-28-01353-f005:**
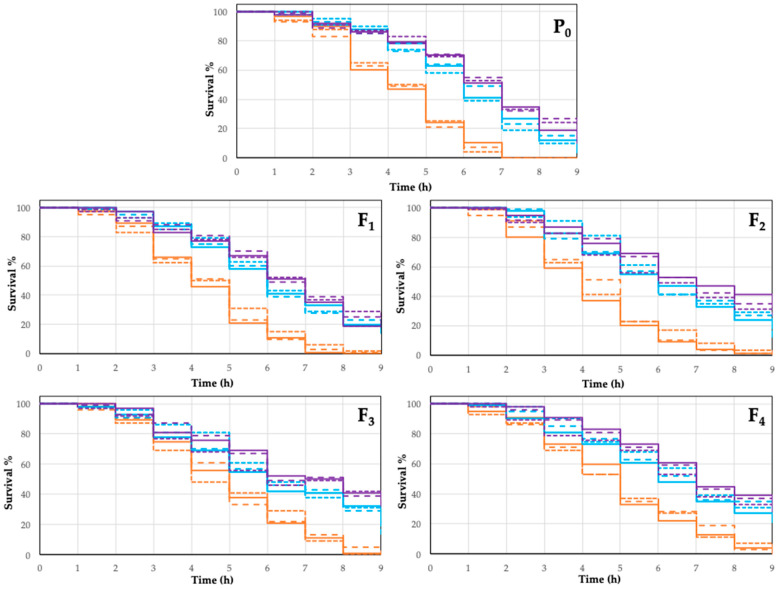
Comparison of the survival rate of *Caenorhabditis elegans* treated with 50 µM anthocyanins extracted from coffee (*Coffea arabica* L.) husks using ethanol as the extracting agent. Fifty µM anthocyanin conjugates from the same extract attached to zinc oxide nanoparticles [fresh (purple) and after 12 weeks (light blue)] and exposed to oxidative stress conditions (10 mM H_2_O_2_). All treatments were compared versus the control group (orange), which did not receive any treatment. The effects of the anthocyanins were analyzed in the parental line (P0) and four offspring generations (F1, F2, F3, and F4). The data shown correspond to three independent studies.

## Data Availability

The data that support the findings of this study are available from the corresponding author upon reasonable request.

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
