# Peer review of "Stabilization of Anthocyanins from Coffee (Coffea arabica L.) Husks and In Vivo Evaluation of Their Antioxidant Activity"

_molecules, 2023, doi:10.3390/molecules28031353_

Round 1

Author Response

Reviewer 1 

Comment 1:

First, I have some minor suggestions:

Title: prefer "husks" instead of "husks;"

Place Table 1 as supplementary material.

Figure 2 is unclear; please separate the HPLC profile (A) and NMR spectrum (B).

Responses to Comment 1:

  • The title was adjusted as suggested.
  • Table 1 has been placed on the Supplementary Material section.
  • 2 has been modified, separating HPLC profile from NMR spectrum.

Comment 2:

Second, I'm curious if color shift occurs in the presence of ZnNPs. Metal chelation of polyphenols, in particular anthocyanins, has been largely studied for commercial applications (e.g., in the wine industry) and appears to greatly influence the color.

Here, for instance, it would be good to show the color appearance of the extract in different extraction conditions (basic solvents (water, ethanol) as well as in the presence of ZnNPs). Providing the UV spectra of these different extracts would also be of great interest to the reader.

I recommend the present work for publication in Molecules after considering these revisions.

Responses to Comment 2:

  • There is no shift observed in the spectra caused for the presence of ZnO nanoparticles nor changes in color were detected (Fig. S3).
  • The color appearance is now shown in Fig. S2, and in Fig. S3 we can observe the spectra of the extracts and the preparations.
  • Thank you for considering this work for publication after these revisions.

Reviewer 2 Report

The present paper introduces a novel way for the extraction and stabilization of anthocyanins from coffee husks, followed by the evaluation of the biological activity using the C. elegans model.

The method seems promising from a scientific point of view, but I'm not sure that commercial applications would be easily obtained because of the use of NPs. However, from a basic science viewpoint, I consider that this paper deserves publication.

First, I have some minor suggestions:

  • Title: prefer "husks" instead of "husks;"
  • Place Table 1 as supplementary material.
  • Figure 2 is unclear; please separate the HPLC profile (A) and NMR spectrum (B).

Second, I'm curious if color shift occurs in the presence of ZnNPs. Metal chelation of polyphenols, in particular anthocyanins, has been largely studied for commercial applications (e.g., in the wine industry) and appears to greatly influence the color.

Here, for instance, it would be good to show the color appearance of the extract in different extraction conditions (basic solvents (water, ethanol) as well as in the presence of ZnNPs). Providing the UV spectra of these different extracts would also be of great interest to the reader.

I recommend the present work for publication in Molecules after considering these revisions.

Author Response

Reviewer 2

Comment 1:

There are some typos errors in the manuscript. Specially the molecular formula of water in text (e.g., pg 5 line 124) and Fig. 1 as well. Please write using the subscript. Define the abbreviations where it comes first not in middle of anywhere.

Responses to Comment 1:

  • The errors were corrected, including the molecular formulas of water.
  • The abbreviations have been defined at the beginning of the document.

Comment 2:

The authors should explain why they choose the ZnO nanoparticles for stabilizing the anthocyanins.

Comment 3:

Please explain why ZnO does not take participate in the reaction when we need only anthocyanin for the purpose? Or how ZnO can be separated from the anthocyanin?

Responses to Comments 2 and 3:

  • The use of ZnO has been argued within the text (lines 114-121).
  • The non-participation of ZnO with anthocyanins has been discussed within the text (lines 114-121).

Comment 4:

Highlight the novelty of this study in the last paragraph.

Response to Comment 4:

  • The novelty of the study has been highlighted in the last paragraph (lines 515-528).

Comment 5:

The authors include the study of AAS in section 3.10. but have not placed the any graph or data in the manuscript. Please include it in the main text.

Response to Comment 5:

  • The AAS data and the related discussion have been included within the text (lines 190-202; Figs. 4 and S1).

Comment 6:

Identify the major limitations of this study and suggest ways to improve them. Given the above-mentioned points, the current version of the manuscript required minor revision before it will be considered for publication in the molecules journal.

Response to Comment 6:

  • The main limitations of the study and improvements have been mentioned within the text (lines 520-526).

Round 2

Reviewer 2 Report

All of my suggestions were taken into account, and all of my questions were perfectly answered.

This manuscript is recommended for publication in Molecules.